# All Ways Lead to Rome—Meiotic Stabilization Can Take Many Routes in Nascent Polyploid Plants

**DOI:** 10.3390/genes13010147

**Published:** 2022-01-15

**Authors:** Adrián Gonzalo

**Affiliations:** Department of Biology, Institute of Molecular Plant Biology, Swiss Federal Institute of Technology (ETH) Zürich, 8092 Zürich, Switzerland; adrian.gonzalo@biol.ethz.ch

**Keywords:** polyploidy, meiosis, evolution

## Abstract

Newly formed polyploids often show extensive meiotic defects, resulting in aneuploid gametes, and thus reduced fertility. However, while many neopolyploids are meiotically unstable, polyploid lineages that survive in nature are generally stable and fertile; thus, those lineages that survive are those that are able to overcome these challenges. Several genes that promote polyploid stabilization are now known in plants, allowing speculation on the evolutionary origin of these meiotic adjustments. Here, I discuss results that show that meiotic stability can be achieved through the differentiation of certain alleles of certain genes between ploidies. These alleles, at least sometimes, seem to arise by novel mutation, while standing variation in either ancestral diploids or related polyploids, from which alleles can introgress, may also contribute. Growing evidence also suggests that the coevolution of multiple interacting genes has contributed to polyploid stabilization, and in allopolyploids, the return of duplicated genes to single copies (genome fractionation) may also play a role in meiotic stabilization. There is also some evidence that epigenetic regulation may be important, which can help explain why some polyploid lineages can partly stabilize quite rapidly.

## 1. Suppression of Meiotic Defects in Polyploids in Plants

Polyploids, which result from whole genome duplication (WGD) events, are thought to have enhanced adaptability, which might explain why they are so pervasive in nature [1]. Their strong stress resilience, e.g., to drought and salinity, and sometimes larger fruits and grains, may explain why they are especially common among crop plants [2]. Some of these changes arise as a direct consequence of the effect genome duplication has on cell size [3]. Polyploids possess multiple advantages that make ploidy an intriguing tool for crop improvement and a key phenomenon in adaptive evolution in nature. However, newly formed (neo)polyploids often also exhibit meiotic irregularities that compromise their genome stability and fertility [4,5,6,7,8,9].

After WGD, the presence of multiple copies of each chromosome causes meiosis to deviate from how it happens in diploids. During the diploid meiotic program, pairs of homologous chromosomes align and pair with each other at one or more points along their length [10]. Nascent recombination events between paired chromosomes initiate the polymerization of the synaptonemal complex (a proteinaceous structure that holds both homologs together across their entire length during the prophase [10]). A subset of recombinational interactions eventually mature into meiotic crossovers [11] that, along with sister chromatid cohesion, physically link homologous chromosomes (cytologically visualized as chiasmata during metaphase I) in pairs called bivalents that later segregate towards opposite poles in anaphase I [12]. In neopolyploids, multiple homologous partners are suddenly present, and promiscuous interactions can arise among them, leading to so-called multivalents (Figure 1a) that can lead to chromosome mis-segregation [4,5,9,13]. 

The pervasiveness of stable and fertile polyploids in both crops and wild plants demonstrates that, after WGD, a specialized meiotic program can evolve that prevents promiscuous interactions and consequent meiotic irregularities [9,14,15]. This process of evolving adjustments to suppress meiotic defects is referred to as cytological diploidization (Figure 1a), because it results in a diploid-like cytologically observable meiosis [16]. The types of adjustment that occur differ by polyploid type, with autopolyploids (originated from within-species WGD) generally evolving bivalent formation in the absence of partner preferences (Figure 1a) [13], and allopolyploids (in which WGD is preceded or followed by inter-specific hybridization), generally evolving crossover preferences among more similar homologs from the same sub-genome (Figure 1a) [17]. Intermediate cases, such as segmental allopolyploids (containing autopolyploid and allopolyploid segments) or autopolyploids with extensive and sustained structural heterozygosity remain less explored [18].

In the last decade, there has been substantial progress in understanding the genic [19,20,21,22,23] and mechanistic [8,9,14,15,24] basis of polyploid stabilization. This work in aggregate has confirmed that the specialized meiotic program differs between auto- and allopolyploids. However, it is unclear whether other potential (and less well studied) meiotic defects, not directly associated to multivalent formation (e.g., repair defects [25], interlocks [26,27], chromosome movements, etc.), are driven by common challenges (e.g., doubled chromosome content, dosage changes, etc.) to both polyploid types. Although the specific defects of polyploids, and how they are corrected, have been discussed elsewhere [13,17,28,29], here I discuss where these genes come from; in other words, the “genetic route” to stable polyploidy.

## 2. Evolutionary Routes to Meiotic Stability in Polyploids

### 2.1. Mutation and Selection on Nucleotide Variation

The evolution of meiotic stability, given its importance for fertility of polyploid lineages, likely occurs quite rapidly. But how? Where do the alleles come from? One common assumption is that novel adaptations, when they arise rapidly, likely arise from pre-existing standing variations, rather than de novo mutation [30].

In fact, there is evidence in some cases that some variations can pre-exist in diploid ancestors that evolution can select upon in the polyploid lineage (Figure 1c) [30,31,32]. What is more surprising, however, is that in many cases, there is evidence that in the well-studied *Arabidopsis arenosa* autopolyploids [30] and *Brassica napus* allotetraploids [33] there are (post-WGD) de novo allelic variants potentially involved in meiotic stability (Figure 1b). How can this happen so fast? At least eight meiosis genes show strong evidence of selection in tetraploid *A. arenosa* [19,34,35], yet the entire lineage seems to be only about 30,000 generations old [36]. Perhaps, relevant to this, is that polyploids are known for faster mutation accumulation rates than diploids, which can be explained by multiple factors, likely acting simultaneously: first, polyploids have a relaxed purifying selection, due to redundant copies of each gene, which reduces the efficiency of selection to counteract recessive mutations [37,38]. This mechanism, which presumably operates both in auto- and allopolyploids [39,40], is likely more prevalent in autopolyploids where each locus segregates four alleles, as opposed to the only partial redundancy allopolyploids enjoy [41]. Indeed, the sheltering of recessive deleterious mutations has been suggested as a causal factor in the accumulation of transposable elements [39], and the buildup of nucleotide variation or genetic load [42] in *A. arenosa* autotetraploids. Moreover, WGD is known to be accompanied by genome rearrangements [43,44,45], possibly driven by illegitimate recombination between regions with accumulated transposable elements [45], or between homoeologous chromosomes (in allopolyploids) [43]. Such genome rearrangements, well studied in allopolyploids, may occur directly after WGD in neopolyploids, but also in better established polyploids [4,5,46,47] and, in turn, drive further homoeologous exchanges [48]. The result of these genome rearrangements includes presence/absence variation [43], generation of novel recombinant transcripts [49], and changes in expression [47].

### 2.2. By Co-Evolution of Multiple Genes

Recent findings support the idea that meiotic stabilization is not generally monogenic. For example, genome scans in *A. arenosa* autotetraploids suggest at least eight interacting meiosis genes were under selection in the polyploids [19,30,34,35]. The functional validation of allelic differentiation of two candidate genes, ASY1 and ASY3, showed these genes clearly have a relevant effect [21] but did not fully recapitulate all the adjustments required to explain the meiotic phenotype of established autotetraploids [9], suggesting the remaining genes under selection are also important. Interestingly, some of the interacting meiotic factors under selection in *A. arenosa* autotetraploids have a pattern of mutation accumulation consistent with co-evolving networks (Figure 1d) [30]. This may well be a common feature of the evolution of modifications to meiosis, since these proteins are all essential for fertility, and thus cannot easily sustain large effect mutations.

### 2.3. By Genome Fractionation

Genome fractionation is a process by which duplicated genes in an allopolyploid return to single copy [50]. During this genomic erosion process, duplicated genes can be lost as a consequence of rearrangements, or pseudogenized by loss-of-function mutations. Given that this genic diploidization occurs in parallel with cytological diploidization starting directly after WGD [16], some authors have speculated these two processes might be entwined [51]. Interestingly, meiotic recombination and DNA repair genes are among the most rapidly returned to single copy [52,53]. This rapid duplicate loss could arise in a neutral way if genes have a very robust dosage-independency, or if positive selection favors dosage reduction of a gene that, when duplicated, can be deleterious for a polyploid (Figure 1e) [54]. Alternatively, it could be that meiosis genes are especially strongly co-evolved, and thus would show biased fractionation towards one parent, or it could be that the heterozygosity of these genes is inherently bad. However, does this fractionation contribute to meiotic stabilization? In keeping with the idea that dosage can be a factor, knocking out one of the two copies carried by *B. napus* of *MSH4*, the meiotic gene with the fastest fractionation pattern observed in eukaryotes, prevents illegitimate crossovers between homoeologous chromosomes in allohaploids while levels of legitimate homologous crossovers are maintained in euploids [51]. This observation suggests that the fractionation of certain genes might indeed be adaptive for allopolyploid meiosis (Figure 1e), though the generality of this pathway remains to be tested.

### 2.4. By Epigenetic Regulation

Interestingly, there is good evidence that the meiotic stability can substantially improve relatively fast within a few generations after inducing WGD, both in neo-autopolyploids such as red clover [55], rye [56,57] or *Arabidopsis thaliana* [58], and synthetic neo-allotetraploids such as *Brassica* trigenomic allohexaploids [59,60], *xBrassicoraphanus* [61], *Tragopogon* [62], triticale [63,64,65] or wheat [66]. In some of these examples it was clear that meiotic stability evolved from little or no standing variation, and so quickly that a de novo mutation route was unlikely: for instance, Santos et al. [58], showed how in *A. thaliana* autopolyploids, generated by colchicine treatment in a homozygous Columbia-0 background, multivalent frequency can be significantly reduced in ten generations of self-fertilization, starting from complete homozygosis. Epigenetics offers a tempting hypothesis that might explain the quick meiotic stabilization of polyploids without preexisting meiotic variability. Recent work has compared the epigenetic and gene expression changes in established *B. napus* and *Arabidopsis suecica* allopolyploids, relative to recently synthesized neo-allopolyploids [7,67,68]. Following this approach, two studies of *A. suecica* [7,68] identified genes that underwent DNA methylation and expression changes through the process of stabilization (Figure 1f,g), though whether this contributes to meiotic stabilization directly remains to be tested. Interestingly, hypo-methylated genes, associated with expression upregulation, were enriched in reproduction-associated and meiotic genes [7,68], including *SMC6B* and *PDS5B*. SMC6B is part of the conserved SMC5/6 complex which is essential for chromosome stability during DNA repair, DNA replication and cell division [69,70]. Intriguingly, recent findings have indicated that the lack of a functional SMC5/6 complex drives much stronger meiotic defects in *A. thaliana* neo-autotetraploids than in diploids [25]. *PDS5B* is one of five PDS5 orthologs present in *A. thaliana* [71], and is a regulator of the cohesion complex during mitosis and meiosis, which in turn controls axis length and crossover numbers during meiosis in mice [72] and budding yeast [40]. These two traits are diminished in established *A. arenosa* autotetraploids compared with neo-autotetraploids, likely contributing to the meiotic stability of this lineage [9], which might explain why PDS5B is under strong selection in *A. arenosa* autotetraploid natural populations [19]. Interestingly, the downregulation of either *SMC6B* or *PDS5* was shown to lead to meiotic instability in allohexaploid wheat [66]. Similar epigenetic changes have been observed in *B. napus* with genes encoding DNA repair proteins involved in meiosis, such as the ligase LIG4 or the mismatch repair MSH6, or NBS1, from the DNA double-strand break-processing complex [67]. Whether the epigenetic regulation of these genes in established polyploids is just coincidental or if, in fact, the epigenetic down- or upregulation of these genes plays a role in meiotic stabilization of polyploids is an exciting question that could be addressed with functional studies.

### 2.5. By Pre-Adaptation

Another quick pathway for meiotic stabilization in neopolyploids can happen post-WGD, when the meiotic stability promoters are selected from the available pre-adapted variants that arose in a related contemporary species or in a past antecessor. These pre-adapted meiotic factors can pass through species horizontally, by gene flow (adaptation via hybridization), or by descent (adaptation from standing variation). In *Arabidopsis lyrata*, genome scans comparing diploid and autotetraploid populations detected selection signatures on the same meiotic genes that are under strong selection in *A. arenosa* tetraploids [42]. While investigating whether these tetraploid-adapted alleles emerged independently or from a common origin, the authors found clear evidence of gene flow from autotetraploid *A. arenosa* to autotetraploid *A. lyrata*. This suggests that pre-adapted meiotic alleles introgressed into *A. lyrata* through inter-specific hybridization with *A. arenosa*, where they were positively selected, presumably contributing to meiotic stability (Figure 1h) [42,73]. The other pathway to meiotic pre-adaptation for WGD is by descent. Recently, it was shown that meiotic stability of *A. arenosa* neo-auto-octoploids generated by whole genome quadruplication of diploids displayed extensive meiotic aberrations, whereas neo-auto-octoploids produced by WGD of an established tetraploid show markedly more regular meiosis [9]. This suggest that meiotic adjustments that stabilize autotetraploid meiosis can pre-adapt a lineage for subsequent WGD events.

## 3. Conclusions and Perspectives

As more loci involved in meiotic stability of polyploids are characterized, more evidence supports the notion that there is diversity in the adaptive solutions evolved. Similarly, these discoveries hint that the evolutionary pathways leading to these meiotic adaptations can be equally diverse. In addition to the classic route of mutation of single genes followed by selection, recent studies—outlined in this review—suggest some other less obvious ways: multiple co-evolving genes, genome fractionation, epigenetic regulation or pre-adaptation. Future discoveries will likely be fueled by new methods to quantify how stable (or unstable) a polyploid is in a precise and high-throughput manner that are not solely dependent on time-extensive cytological methods (e.g., Higgins et al. 2021 [22]). Furthermore, a better understanding on how meiotic stability evolves in polyploids might inspire strategies to artificially stabilize synthetic polyploids (e.g., knocking out copies of particular duplicated genes, introgression between species, generation of epialleles, etc.).

## Figures and Tables

**Figure 1 genes-13-00147-f001:**
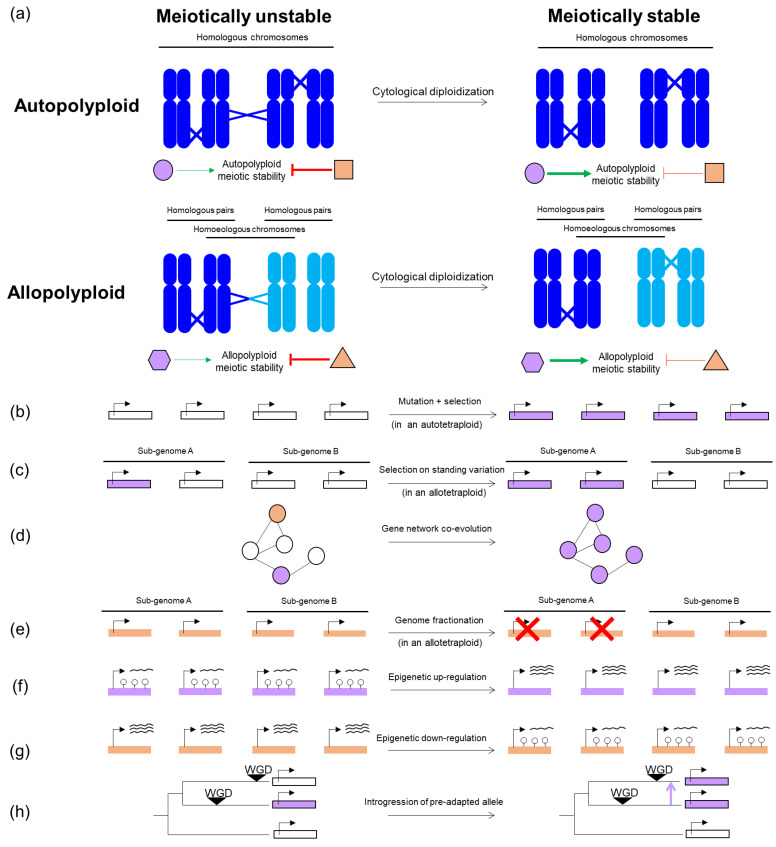
Mechanisms (**a**) and evolutionary routes (**b**–**h**) to meiotic stabilization in auto- and allopolyploids. Genes and factors that promote or counteract polyploid meiotic stability are shown in violet or orange, respectively, while genes or factors with undetermined effect on meiotic stability are shown in white. Different set of shapes are used to represent auto- and allopolyploid factors (circle/square and hexagon/triangle, respectively) to indicate that those factors are not expected to coincide for the two kinds of polyploids. (**a**) Schematic representation of a putative tetraploid organism (4x = 2n = 4), both as an autopolyploid and allopolyploid, before and after undergoing cytological diploidization (with and without crossovers involving more than two chromosomes, respectively). Homologous chromosomes from the same sub-genome are shown in the same color (either dark or light blue), while homoeologs are shown in different color (different sub-genomes). (**b**–**h**) Schematic representations of putative genes (rectangles) and factors (shapes including circles, squares, hexagons or triangles) taking different routes to contribute to meiotic stabilization. Genes are shown in four copies to represent the four possible alleles that can coexist in a tetraploid. When relevant for allopolyploidy cases, two different sub-genomes are represented.

## Data Availability

Not applicable.

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
