# Peer review of "All Ways Lead to Rome—Meiotic Stabilization Can Take Many Routes in Nascent Polyploid Plants"

_genes, 2022, doi:10.3390/genes13010147_

Round 1
Reviewer 1 Report
The manuscript of Gonzalo A. is devoted to the meiotic stability achievement in neopolyploid species. I would recommend clarifying in the title that all mentioned studies were done on plants. Otherwise, it would be better to refer to the existing researches on neopolyploid studies in animals. Please, find below the comments and suggestions on the manuscript.
L. 9-10. 'can lead to aneuploidy and low fertility' - I would recommend rephrasing here.
L.12-13. are known in plants.
L. 25. in plants.
L. 59-60. What about cases of autopolyploids with the sustained heterozygosity of the genome?
L. 75-76. autopolyploid A. arenosa autopolyploids?
L. 76. Post-WGD→post-WGD
L. 82 First→first
L. 88. The burst of TE activity occurring in allopolyploids after WGD usually lead to significant genome re-shuffling.
L. 90. add a reference (genome rearrangements)
L.90-92. what about chromosome rearrangements in neopolyploids? for instance, massive chromosome fusions or chromosome number reduction.
L. 116. It→it
L. 132. For→for
Author Response
Dear editor and reviewers,
I truly appreciate your fair feedback and comments. Therefore, I have made all the proposed changes which were very appropriate. Here I detail all those edits. Please keep in mind that some of the edits required new references that I have also added to the Bibliography.
Sincerely yours,
Adrian
Reviewer 1
The manuscript of Gonzalo A. is devoted to the meiotic stability achievement in neopolyploid species. I would recommend clarifying in the title that all mentioned studies were done on plants. Otherwise, it would be better to refer to the existing researches on neopolyploid studies in animals. Please, find below the comments and suggestions on the manuscript.
Thank you for this point. To address this fair concern, I have changed the tittle accordingly and done the changes suggested in lines 9-10, 12-1, and 25.
Tittle: All ways lead to Rome – meiotic stabilization can take many routes in nascent polyploids plants
- 9-10. 'can lead to aneuploidy and low fertility' - I would recommend rephrasing here.
Done. Now, the corrected sentence is: “Newly formed polyploids often show extensive meiotic defects resulting in aneuploid gametes and thus reduced fertility”
L.12-13. are known in plants.
Done.
- 25. in plants.
Done. (now in line 26)
- 59-60. What about cases of autopolyploids with the sustained heterozygosity of the genome?
To acknowledge these cases I have included this sentence: “Intermediate cases like segmental allopolyploids (containing autopolyploid and allopolyploid segments) or autopolyploids with extensive and sustained heterozygosity remain less explored [18]”
- 75-76. autopolyploid A. arenosa autopolyploids?
Done.
- 76. Post-WGD→post-WGD
Done (now in line 79).
- 82 First→first
done
- 88. The burst of TE activity occurring in allopolyploids after WGD usually lead to significant genome re-shuffling.
To address this comment and those pointed out in lines 90 and 90-92. I have slightly modified this part of the text and including some additional references as suggested. Please see current lines 93-99:
“Moreover, WGD is known to be accompanied by boosts in certain mutagenic events such as genome rearrangements[42–44], possibly driven by non-allelic recombination between accumulated transposable elements [44], or homoeologous chromosomes (in allopolyploids) [42]. Such genome rearrangements, well studied in allopolyploids, may happen right after WGD in neopolyploids, but also in better established polyploids [4,5,45,46] and These, are well studied in allopolyploids, where they in turn drive fur-ther homoeologous exchanges [47]. The result of these genome rearrangements in-cludes that mediate presence/absence variation [42], generation of novel recombinant transcripts [48], and changes in expression [46].”
- 90. add a reference (genome rearrangements)
Done (see paragraph above)
L.90-92. what about chromosome rearrangements in neopolyploids? for instance, massive chromosome fusions or chromosome number reduction.
Done (see paragraph above)
- 116. It→it
Done (now in line 124)
- 132. For→for
Done (now in line 137)
Additional minor edits:
While making the changes suggested by the reviewers, I also made a few minor edits that improved the text in my opinion.
L.33 redundant intriguing”
- 40. SC abbreviation spelled out (Synaptonemal complex)
- 60-61. Redundancy “genic” basis and “genetic control”
- 78 at à in
L.100. à I rephrased the section title for the sake of consistency with sections bellow
L.123-124. Rephrasing to avoid ambiguity (duplicated gene, not duplicated polyploid)
L144-149. One sentence removed to avoid redundancy.
- 159. Redundant “in in”
L.169-170. DSB abbreviation spelled out (double strand breaks)
L.208-210. Added more precise information about funding.

Reviewer 2 Report
This is well written Review, interesting not only for plant genetisists but for a wider audience studying polyploidization processes.
Author Response
Dear editor and reviewers,
I truly appreciate your fair feedback and comments. Therefore, I have made all the proposed changes which were very appropriate. Here I detail all those edits. Please keep in mind that some of the edits required new references that I have also added to the Bibliography.
Sincerely yours,
Adrian
Additional minor edits:
While making the changes suggested by the reviewers, I also made a few minor edits that improved the text in my opinion.
L.33 redundant intriguing”
- 40. SC abbreviation spelled out (Synaptonemal complex)
- 60-61. Redundancy “genic” basis and “genetic control”
- 78 at à in
L.100. à I rephrased the section title for the sake of consistency with sections bellow
L.123-124. Rephrasing to avoid ambiguity (duplicated gene, not duplicated polyploid)
L144-149. One sentence removed to avoid redundancy.
- 159. Redundant “in in”
L.169-170. DSB abbreviation spelled out (double strand breaks)
L.208-210. Added more precise information about funding.